

# Randomized feasibility trial of the Scleroderma Patient-centered Intervention Network hand exercise program (SPIN-HAND)

Linda Kwakkenbos[1], Marie-Eve Carrier[2], Joep Welling[3], Kimberly A. Turner[2], Julie Cumin[2], Mia Pépin[2], Cornelia van den Ende[4], Anne A. Schouffoer[5,6], Marie Hudson[2,7], Ward van Breda[8], Maureen Sauve[9,10], Maureen D. Mayes[11], Vanessa L. Malcarne[12], Warren R. Nielson[13], Christelle Nguyen[14], Isabelle Boutron[15,16], François Rannou[14,17], Brett D. Thombs[2,7,18,19,20,21], Luc Mouthon[22,23] and The SPIN Investigators

[1] Radboud University Nijmegen, Nijmegen, Netherlands
[2] Lady Davis Institute of the Jewish General Hospital, Montreal, Canada
[3] NVLE Dutch patient organization for systemic autoimmune diseases, Utrecht, The Netherlands
[4] Sint Maartenskliniek, Nijmegen, The Netherlands
[5] Leiden University Medical Center, Leiden, The Netherlands
[6] Haga Teaching Hospital, The Hague, The Netherlands
[7] Department of Medicine, McGill University, Montreal, Quebec, Canada
[8] Faculty of Behavioural and Movement Sciences, Vrije Universiteit, Amsterdam, The Netherlands
[9] Scleroderma Society of Ontario, Hamilton, Ontario, Canada
[10] Scleroderma Canada, Ottawa, Ontario, Canada
[11] University of Texas McGovern School of Medicine, Houston, Texas, United States of America
[12] San Diego State University, San Diego, CA, United States of America
[13] St. Joseph's Health Care, London, Ontario, Canada
[14] Université de Paris, Paris, France
[15] Centre d'Épidémiologie Clinique, Assistance Publique–Hôpitaux de Paris (AP-HP), Hôpital Hôtel Dieu, Paris, France
[16] Centre of Research Epidemiology and Statistics (CRESS), Inserm, INRA, Université de Paris, Paris, France
[17] Assistance Publique-Hôpitaux de Paris, Paris, France
[18] Department of Psychiatry, McGill University, Montreal, Quebec, Canada
[19] Department of Epidemiology, Biostatistics, and Occupational Health, McGill University, Montreal, Quebec, Canada
[20] Department of Psychology, McGill University, Montreal, Quebec, Canada
[21] Department of Educational and Counselling Psychology, McGill University, Montreal, Quebec, Canada
[22] Service de Médecine Interne, Centre de Référence Maladies Autoimmunes Systémiques Rares d'Ile de France, Hôpital Cochin, Assistance Publique-Hôpitaux de Paris (AP-HP), Paris, France
[23] APHP-CUP, Hôpital Cochin, Paris, France

Corresponding author
Linda Kwakkenbos,
l.kwakkenbos@psych.ru.nl

## ABSTRACT

**Purpose.** The Scleroderma Patient-centered Intervention Network (SPIN) online hand exercise program (SPIN-HAND), is an online self-help program of hand exercises designed to improve hand function for people with scleroderma. The objective of this feasibility trial was to evaluate aspects of feasibility for conducting a full-scale randomized controlled trial of the SPIN-HAND program.

**Materials and Methods.** The feasibility trial was embedded in the SPIN cohort and utilized the cohort multiple randomized controlled trial (cmRCT) design. In the
cmRCT design, at the time of cohort enrollment, cohort participants consent to be assessed for trial eligibility and randomized prior to being informed about trials conducted using the cohort. When trials were conducted in the cohort, participants randomized to the intervention were informed and consented to access the intervention. Participants randomized to control were not informed that they have not received an intervention. All participants eligible and randomized to participate in the trial were included in analyses on an intent-to-treat basis. Cohort participants with a Cochin Hand Function Scale score $\geq$ 3/90 and an interest in using an online hand-exercise intervention were randomized (1:1 ratio) to be offered as usual care plus the SPIN-HAND Program or usual care for 3 months. User satisfaction was assessed with semi-structured interviews.

**Results.** Of the 40 randomized participants, 24 were allocated to SPIN-HAND and 16 to usual care. Of 24 participants randomized to be offered SPIN-HAND, 15 (63%) consented to use the program. Usage of SPIN-HAND content among the 15 participants who consented to use the program was low; only five (33%) logged in more than twice. Participants found the content relevant and easy to understand (satisfaction rating 8.5/10, N = 6). Automated eligibility and randomization procedures via the SPIN Cohort platform functioned properly. The required technical support was minimal.

**Conclusions.** Trial methodology functioned as designed, and the SPIN-HAND Program was feasibly delivered; however, the acceptance of the offer and use of program content among accepters were low. Adjustments to information provided to potential participants will be implemented in the full-scale SPIN-HAND trial to attempt to increase offer acceptance.

**Subjects** Clinical Trials, Dermatology, Drugs and Devices, Kinesiology, Rheumatology

**Keywords** Feasibility trial, Scleroderma, Systemic sclerosis, Tele-rehabilitation, Internet intervention, Physical therapy, Occupational therapy, Cohort multiple RCT

# INTRODUCTION

Patient-centered care emphasizes the need for services and interventions to help manage the psychosocial, educational, and functional aspects of living with chronic diseases, in addition to core medical treatment (*Rathert, Wyrwich & Boren, 2013*; *Stewart, 2001*; *Bokhour et al., 2018*; *Gusmano, Maschke & MZ, 2019*; *Ells, Hunt & Evans, 2011*; *American College of Rheumatology Association of Rheumatology Health Professionals, 2021*). Interventions to support health-related quality of life (HRQL) are often available for patients with common diseases. Patients with rare diseases often face unique challenges that are not addressed by generic interventions or interventions developed for more common conditions; however, there are barriers to developing, testing and disseminating interventions in rare diseases, (*Kwakkenbos et al., 2013*) and access is limited (*Kole & Faurisson, 0000*; *Huyard, 2009*; *Von der Lippe, Diesen & Feragen, 2017*; *Dharssi et al., 2017*).

Systemic sclerosis (SSc, or scleroderma) is a rare systemic autoimmune disease that affects the skin and musculoskeletal system, as well as internal organs including lungs, gastrointestinal tract and cardiovascular system (*Allanore et al., 2015*; *Wigley & Hummers, 2003*). The disease is associated with limitations in physical mobility and hand function,

pain, fatigue, sleep disturbance, sexual dysfunction, and mental health challenges, including body image concerns due to changes in appearance (*Kwakkenbos et al., 2015*; *Bassel et al., 2011*; *Thombs et al., 2010*; *Jewett, Haythornthwaite & Thombs, 2012*). It, thus, has far-reaching consequences for physical health, as well as emotional and social well-being.

Non-pharmacological interventions, including rehabilitation interventions, could potentially help SSc patients to achieve better physical function and HRQL. There is limited evidence for these types of interventions in SSc, however, and 2017 EULAR guidelines for the SSc management did not make recommendations on any rehabilitation or other non-pharmacological interventions due to the lack of evidence on their effectiveness (*Pellar & Pope, 2017*; *Kowal-Bielecka et al., 2009*; *Kowal-Bielecka et al., 2017*). The British Society for Rheumatology/British Health Professionals in Rheumatology guidelines from 2016 similarly indicate that "although the evidence base is limited, non-drug interventions may have merit and are well tolerated" and call for more research (*Denton et al., 2016*).

The Scleroderma Patient-centered Intervention Network (SPIN) was established to work with people with SSc to identify care needs and priorities in order to develop, test and disseminate accessible non-pharmacological interventions that improve HRQL and empower people with SSc to more effectively cope with the disease (*Kwakkenbos et al., 2013*). SPIN is a collaboration of researchers, clinicians, patient organizations, and people with SSc from Canada, the USA, Europe, Latin America and Australia (http://www.spinsclero.com). Over 2,000 SSc patients from 45 centres have been enrolled in SPIN's web-based cohort, and SPIN investigators are developing a series of online interventions to be tested with SPIN Cohort participants in pragmatic randomized controlled trials (RCTs) (*Kwakkenbos et al., 2013*; *Carrier et al., 2018*).

Impaired hand function from contractures, deformities and digital ulcers of the hand are common and a major contributor to overall disability and reduced HRQL in SSc (*Kallen et al., 2010*; *Rannou et al., 2007*; *Kwakkenbos et al., 2018*). Based on SSc patient priorities, SPIN developed an online program of hand exercises designed to improve or reduce deterioration of hand function (SPIN-HAND) (*Carrier et al., 2018*). Previously, three RCTs evaluated physical or occupational therapy interventions that targeted improving hand function in SSc, but all were limited by small samples (N <20 per arm) (*Poole, 2010*). More recently, a trial of a 12-week multidisciplinary day treatment program, which included hand exercises, reported improved grip strength (mean difference between groups 3.0 kg, 95% CI [0.1–5.9]) and improved hand mobility (Hand Mobility In Scleroderma (HAMIS) scale (*Sandqvist & Eklund, 2000*) mean difference between groups $-1.8$, 95% CI [$-3.5$ to $-0.1$]) compared to usual care 24 weeks post-randomization ($N = 53$) (*Schouffoer et al., 2011*). Another trial ($N = 44$) (*Filippetti et al., 2020*) tested the effectiveness of a 6-month, minimally supervised home rehabilitation program that included aerobic exercise, muscular endurance training of the upper limb and stretching exercises of the hands. The authors reported that compared to the control group that received generic recommendations to improve physical exercise, there were no differences in HAMIS scores at 3 and 6 months after the start of the intervention. Finally, a recent RCT tested a one-month supervised SSc home-based exercise therapy program, which included hand exercises ($N = 218$) (*Rannou et al., 2017*). The program reduced hand disability significantly at 1-month follow-up

(Cochin Hand Function Scale [CHFS] (*Duruöz et al., 1996*) adjusted mean difference −3.65, 95% CI [−6.12 to −1.17]), but there was not a difference between groups at 6-months (CHFS adjusted mean difference −0.8, 95% CI [−3.6–2.0]) or 12-months post-randomization (CHFS adjusted mean difference 0.5, 95% CI [−3.1–4.0]).

Prior to conducting a full-scale RCT to assess the effectiveness of offering the SPIN-HAND program to improve hand function compared to care as usual, we assessed the feasibility of the planned trial methodology and the user-friendliness and acceptability of the SPIN-HAND online program in a randomized feasibility trial (*Thabane et al., 2010*; *Van Teijlingen et al., 2001*; *Kraemer et al., 2006*). The aim of the SPIN-HAND Feasibility Trial was to inform adjustments to the online intervention and the trial protocol for the full-scale SPIN-HAND Trial, by obtaining data related to the study's process, required resources and management, scientific aspects, and participant acceptability of the enrolment procedures and SPIN-HAND program content (*Carrier et al., 2018*).

## METHODS

### Design and setting

We conducted a parallel-arm, multi-centre feasibility RCT of the online SPIN-HAND Program. The feasibility trial was embedded in the SPIN Cohort, as described in the published trial protocol (*Carrier et al., 2018*). It was registered prior to enrolling participants (NCT03092024) and is reported based on items from the CONSORT extension for Randomised Pilot and Feasibility Trials (*Eldridge et al., 2016*) and the CONSORT extension for Trials Conducted Using Cohorts and Routinely Collected Data (CONSORT-ROUTINE) (*Kwakkenbos et al., 2021*). There were no changes to the feasibility trial protocol after commencement of the trial.

The SPIN Cohort was developed as a framework for embedded pragmatic trials using the cohort multiple RCT (cmRCT) design (*Kwakkenbos et al., 2013*; *Relton et al., 2010*). Participants in the SPIN Cohort enroll in an observational cohort with regular outcome measurement and consent to (1) allow their data to be used for observational studies; (2) allow their data to be used to assess intervention trial eligibility and, if eligible, be randomized; (3) if randomized to the intervention arm of a trial to be contacted by SPIN personnel to be invited to participate in a SPIN intervention; and (4) if randomized to usual care, use their data to evaluate intervention effectiveness without being notified that they have been randomized to the usual care group and not offered the intervention (*Kwakkenbos et al., 2013*). Participants randomized to the intervention arm must consent to access the intervention but are included in the trial arm whether or not they consent to intervention access. Participants in the control arm are not notified about the trial but continue to provide outcome data as part of normal cohort assessments. In the cmRCT design, all participants who are randomized to a trial arm are analyzed on an intent-to-treat basis.

The SPIN Cohort study was approved by the Research Ethics Committee of the Centre intégré universitaire de santé et de services sociaux du Centre-Ouest-de-l'Île-de-Montréal (#MP-05-2013-150) and by the research ethics committees of each participating centre.

Ethics approval for the SPIN-HAND Feasibility Trial was obtained from the Research Ethics Committee of the Centre intégré universitaire de santé et de services sociaux du Centre-Ouest-de-l'Île-de-Montréal (#2019-1146).

## SPIN Cohort participants

To be eligible for the SPIN Cohort, patients needed to be classified as having SSc based on 2013 ACR/EULAR criteria (*Van den Hoogen et al., 2013*) confirmed by a SPIN physician, be ≥18 years old, be able to give informed consent, be fluent in English, French or Spanish, and be able to respond to questionnaires via the internet. The SPIN Cohort is a convenience sample (*Kwakkenbos et al., 2013*; *Carrier et al., 2018*). Eligible SPIN Cohort patients are recruited at SPIN sites (https://spinsclero.com/en/cohort/sites) during regular medical visits, and written informed consent is obtained. A medical data form is submitted online by the site to enrol participants. Cohort participants complete outcome measures via the internet upon enrolment and subsequently every 3 months (*Kwakkenbos et al., 2013*). SPIN Cohort enrollment started in March 2014 and is ongoing. Characteristics of participants in the SPIN Cohort are comparable to those of participants in other large SSc cohorts (*Dougherty et al., 2018*).

## SPIN-HAND feasibility trial participants

Assessment of trial eligibility occurred during participants' regular online SPIN Cohort assessments. Cohort participants were eligible for the SPIN-HAND feasibility trial if they completed their SPIN Cohort measures in English, reported at least mild hand function limitations (CHFS (*Duruöz et al., 1996*) ≥ 3), and indicated interest in using an online hand exercise intervention (≥ 7 on 0–10 scale). The full-scale trial will also include patients who complete their Cohort assessments in French.

## Procedure: randomization, allocation concealment, consent and blinding

Randomization with a 1:1 ratio to be offered versus not offered the SPIN-HAND intervention occurred at the time of Cohort participants' regular SPIN Cohort assessments. Eligible Cohort participants, based on questionnaire responses, were randomized automatically as they completed their regular SPIN Cohort assessments using a feature in the SPIN Cohort platform, which provided immediate centralized randomization and, thus, complete allocation sequence concealment (*Carrier et al., 2018*).

Participants randomized to be offered the intervention received an automated email invitation including a link to the SPIN-HAND Program site and the consent form for the intervention. At initial login, they were prompted to consent to participate in the SPIN-HAND Feasibility Trial by verifying agreement with consent elements and providing their email address as the signature. Participants who consented were then re-directed to the introduction page of the SPIN-HAND Program, while patients who logged out before agreeing to the terms of the consent form returned to the consent page upon subsequent logins. A protocol was created to standardize contact attempts via phone and email to participants in the intervention arm. Within the first 10 days after the SPIN-HAND invitation email was sent, SPIN personnel attempted to contact participants

up to a maximum of 6 times to offer them more information about the study, to answer any questions, and to help them consent or log in to the program. The first attempt was made within 48 h of sending the invitation. If a participant was still unreachable after five attempts, the sixth and last attempt was conducted by SPIN personnel to complete the call protocol at approximately 20 days post-invite. Participants who accepted the offer to use the SPIN-HAND intervention could use the web link to enter the secure intervention site. Email and phone technical support were available during office hours to help participants with the consent process and to access and use the intervention site (*Carrier et al., 2018*).

Participants who were randomized to care as usual were not notified that they had not been offered the intervention and completed their regular SPIN Cohort assessments (*Kwakkenbos et al., 2013*). Thus, participants who were offered the intervention were not blind to their status, whereas participants assigned to usual care were blind to their participation in the trial and to their assignment to usual care (*Carrier et al., 2018*).

## Intervention

SPIN's hand exercise program (*Carrier et al., 2018*) is based on rehabilitation programs that have been shown to improve hand function in SSc (*Rannou et al., 2017*) and rheumatoid arthritis (*Lamb et al., 2015*). It integrates key components of successful disease self-management programs, including goal-setting and feedback, social modeling, and mastery experiences (*Foster et al., 2007*; *Lorig & Holman, 2003*; *Holman & Lorig, 1992*; *Marks, Allegrante & Lorig, 2005a*; *Marks, Allegrante & Lorig, 2005b*). The core of the program consists of four modules that address specific aspects of hand function, including (1) Thumb Flexibility and Strength (3 exercises); (2) Finger Bending (3 exercises); (3) Finger Extension (3 exercises); and (4) Wrist Flexibility and Strength (2 exercises). Participants can select the modules in the order that they prefer, based on a description of the type of function that the module is targeting to maintain or improve (Fig. 1) (*Carrier et al., 2018*). The program is self-administered using the online modules and includes sections on developing a personalized program, goal-setting strategies and examples, progress tracking, sharing goals and progress with friends and family, and patient stories of experiences with hand disability and hand exercises (Figs. 2 and 3) (*Carrier et al., 2018*).

The program utilizes an engaging and easy to navigate web interface. Instructional videos with SSc patients demonstrate and explain how to perform each exercise properly, and additional pictures illustrate common mistakes (Fig. 4). Separate versions of each exercise are available for participants with mild to moderate and more severe hand involvement. Participants can identify their level by reviewing photos of hands of patients with mild to moderate versus more severe involvement (Fig. 5). Some exercise videos include pictures to illustrate alternate versions on how to perform the exercise when there is very severe hand involvement (Fig. 6) (*Carrier et al., 2018*). There are also illustrations of common mistakes in doing the exercises. Patients who use the program are provided guidance on selecting intervention intensity levels and in developing a program that works best for them. Participants offered the program were able to access it for the entire 3-month trial period. They can spend as little or as much time as they wish on individual modules. The program

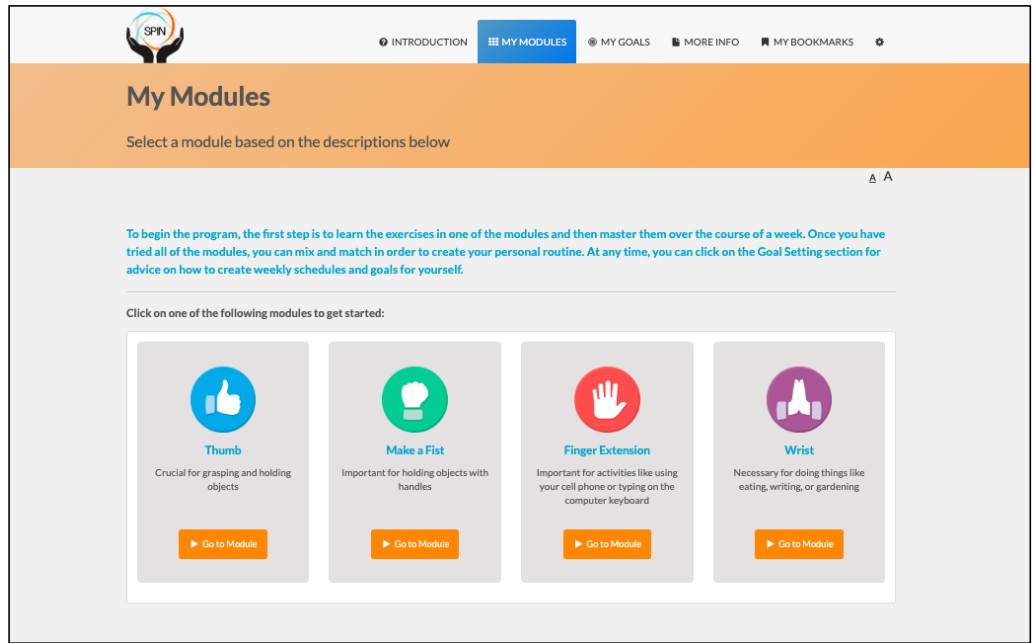

**Figure 1  Menu of the SPIN-HAND Program exercise modules.**

is now available free-of-charge online (https://spinsclero.com/en/toolkits; registration required).

Participants assigned to the care as usual control condition were not notified about the trial and continued to receive their usual health care.

## Feasibility outcomes

Feasibility outcomes that were collected included: (1) the proportion of SPIN Cohort participants who met cut-off thresholds for eligibility during the period of enrollment; (2) the proper functioning of automated eligibility and randomization procedures (*i.e.*, no ineligible participants should be enrolled, and eligible participants based on questionnaire scores should not be missed for enrolment); (3) the proportion of eligible participants randomized to be offered the SPIN-HAND intervention who accepted the offer and consented to participate; (4) rate of completion of trial outcome variables at 3-month follow-up; (5) completeness of the intervention usage log data; (6) ability to successfully link data coming from the SPIN Cohort and SPIN-HAND platforms; (7) trial personnel time requirements to call enrolled participants and help them to consent and use the SPIN-HAND program; (8) any other challenges for study personnel; and (9) technological performance of the online SPIN-HAND program (*Carrier et al., 2018*).

Usage of the SPIN-HAND program modules among participants in the intervention arm were examined via data collected on the number of logins, number of modules accessed, goals set, and time spent on each webpage. At 3-months post-randomization, qualitative semi-structured interviews were conducted with participants in the intervention arm to assess user acceptability and satisfaction. The interview, consisting of 29 questions, was
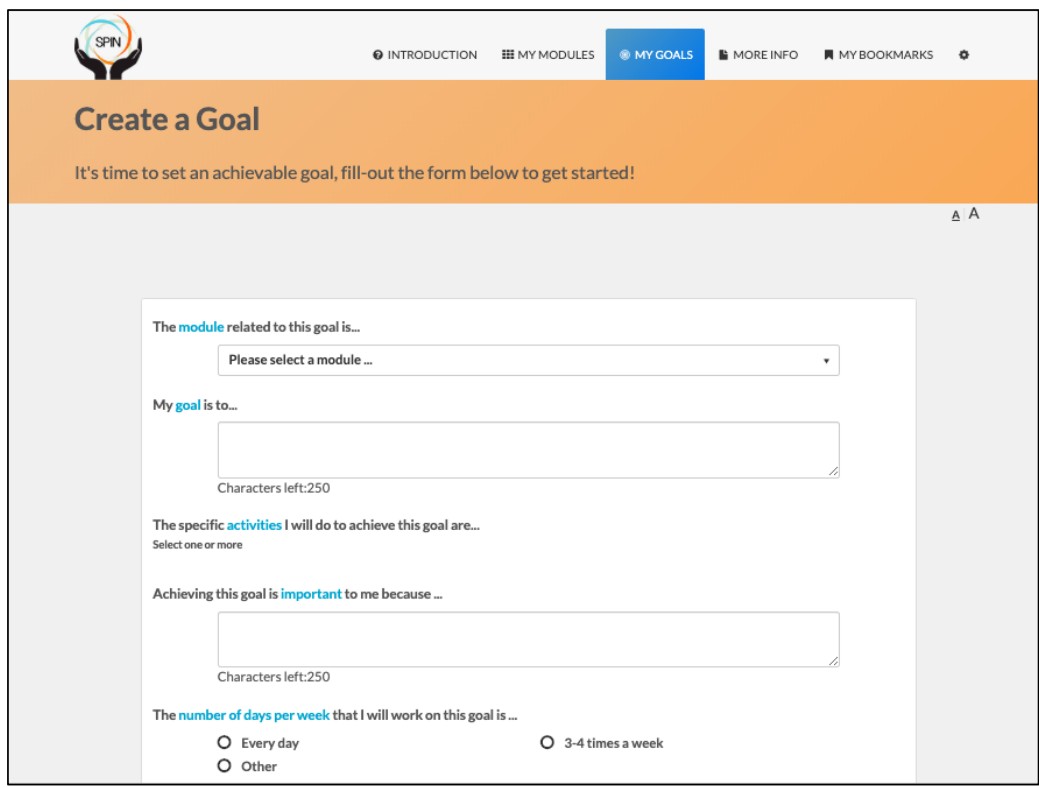

**Figure 2  Goal setting tool.**

guided by items of the Patient Education Materials Assessment Tool for Audiovisual Materials (*Shoemaker, Wolf & Brach, 2014*) and addressed topics related to usability, understandability, organization, and clarity. See Appendix S1A for the interview guide.

## SPIN-HAND planned trial outcome measures

The objectives of the full-scale SPIN-HAND trial will be to evaluate the effect of being offered access to SPIN's online hand exercise program, compared to usual care alone, on hand function (primary), functional health outcomes, and HRQL. Trial outcome measures are routinely assessed as part of the SPIN Cohort assessments every 3-months. Thus, no assessments were added for the trial that were not already part of routine SPIN Cohort assessments. The primary outcome for the full-scale SPIN-HAND trial will be hand function limitations, which will be evaluated using the CHFS (*Rannou et al., 2017*). Secondary outcomes will be patient-reported health status measured with the Patient Reported Outcomes Measurement Information System (PROMIS-29) profile version 2.0 (*The NIH Patient-Reported Outcomes Measurement Information System, 2021*) and HRQL measured with the EuroQoL-5D-5L (EQ-5D) (*Rabin & de Charro, 2001*). Trial outcome measures were collected in the feasibility trial for assessing aspects of feasibility and descriptive purpose. They were not used to evaluate the effectiveness of the SPIN-HAND
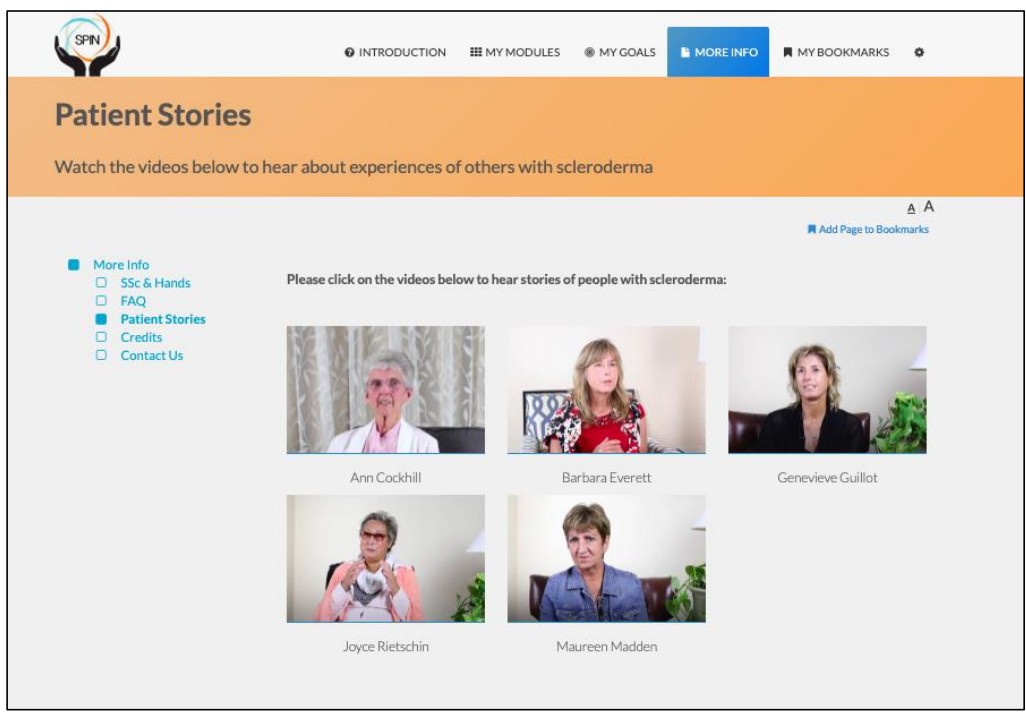

**Figure 3** Patient stories page.

Program. The SPIN-HAND Feasibility Trial was not designed for hypothesis testing or effect size estimation, and the sample size was not appropriate (*Carrier et al., 2018*).

The 18-item CHFS (*Duruöz et al., 1996*) was developed to measure hand function limitations among patients with rheumatic diseases. The CHFS assesses ability to perform hand-related activities (*e.g.*, kitchen, dressing oneself, hygiene, writing/typing). Items are scored on a 0–5 Likert scale (0=*without difficulty;* 5=*impossible*). Higher scores indicate less functionality. The total score is obtained by adding the scores of all items (range 0–90). The CHFS has good convergent validity with general functional disability measures and good sensitivity to change (*Duruöz et al., 1996*; *Poiraudeau et al., 2001*; *Brower & Poole, 2004*). It has been validated in SSc (*Brower & Poole, 2004*).

The PROMIS-29v2 measures seven domains of health status with four items each (physical function, anxiety, depression, fatigue, sleep disturbance, social roles and activities, pain interference) plus a single item for pain intensity (*The NIH Patient-Reported Outcomes Measurement Information System, 2021*). Domain items are scored on a 5-point scale (range 1–5), with different response options for different domains, whereas the single pain intensity item is measured on an 11-point rating scale. Higher scores represent more of the domain being measured; that is, better physical function and ability to participate in social roles and activities, but higher levels of anxiety, depression, fatigue, sleep disturbance, pain interference, and pain intensity. Total raw scores are obtained by summing item scores for each domain, which are converted into T-scores standardized from the general US

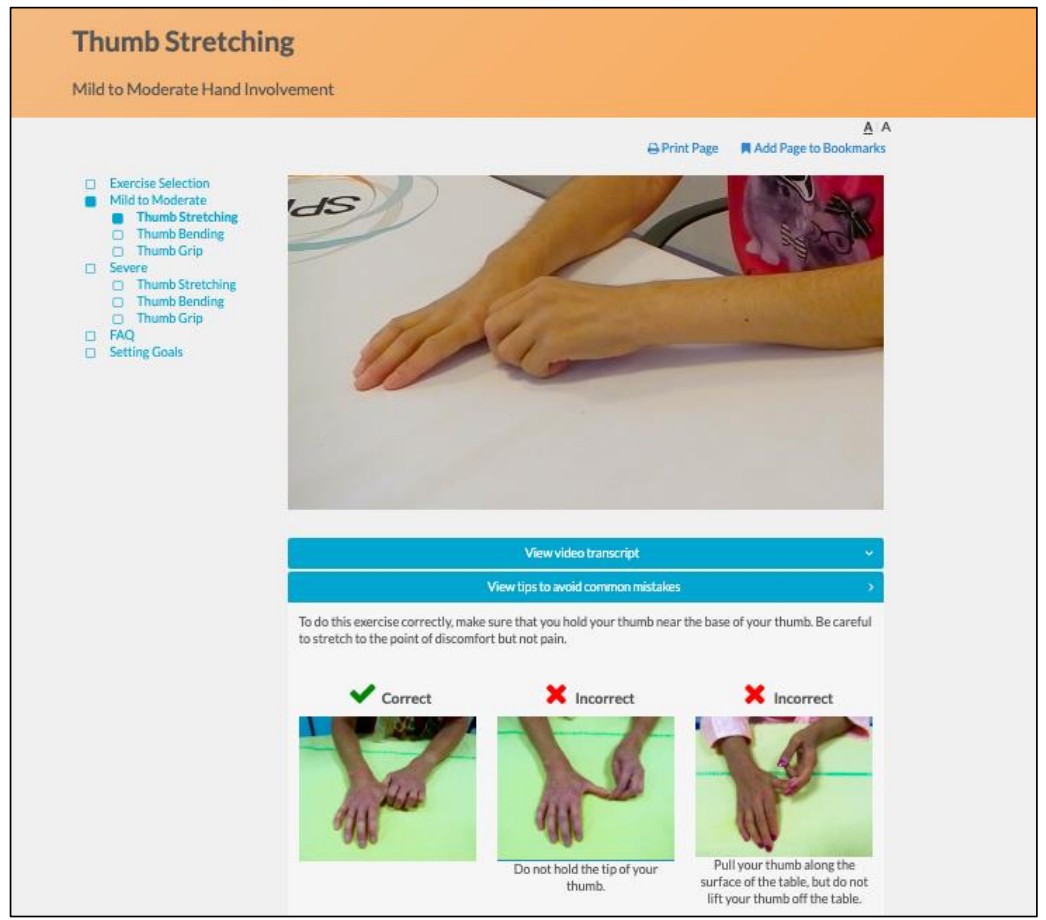

**Figure 4   Instructional video of hand exercise and illustration of common mistakes.**

population (mean = 50, SD = 10). PROMIS-29 version 2.0 has been validated in SSc (*Hinchcliff et al., 2011*; *Kwakkenbos et al., 2017*).

The EQ-5D-5L (*Rabin & de Charro, 2001*) is a 5-item standardized questionnaire, measuring mobility, self-care, usual activities, pain/discomfort, and anxiety/depression. The items are rated from 1 (no problems) to 5 (extreme problems). In addition, a single Visual Analogue Scale records the patient's self-rated health on a scale from 0-100 where the endpoints are labelled *the best health you can imagine* (100) and *the worst health you can imagine* (0).

## Sample size

Since this was a feasibility study, a sample size calculation was not performed. Guidance on the appropriate sample size for feasibility trials varies substantially in the published literature, with rules-of-thumb varying from $N = 12$ to $N = 30$ or more per trial arm (*Sim & Lewis, 2012*; *Julious, 2005*). To ensure that we collected sufficient quantitative and qualitative outcome data to meet our feasibility objectives and inform us about the

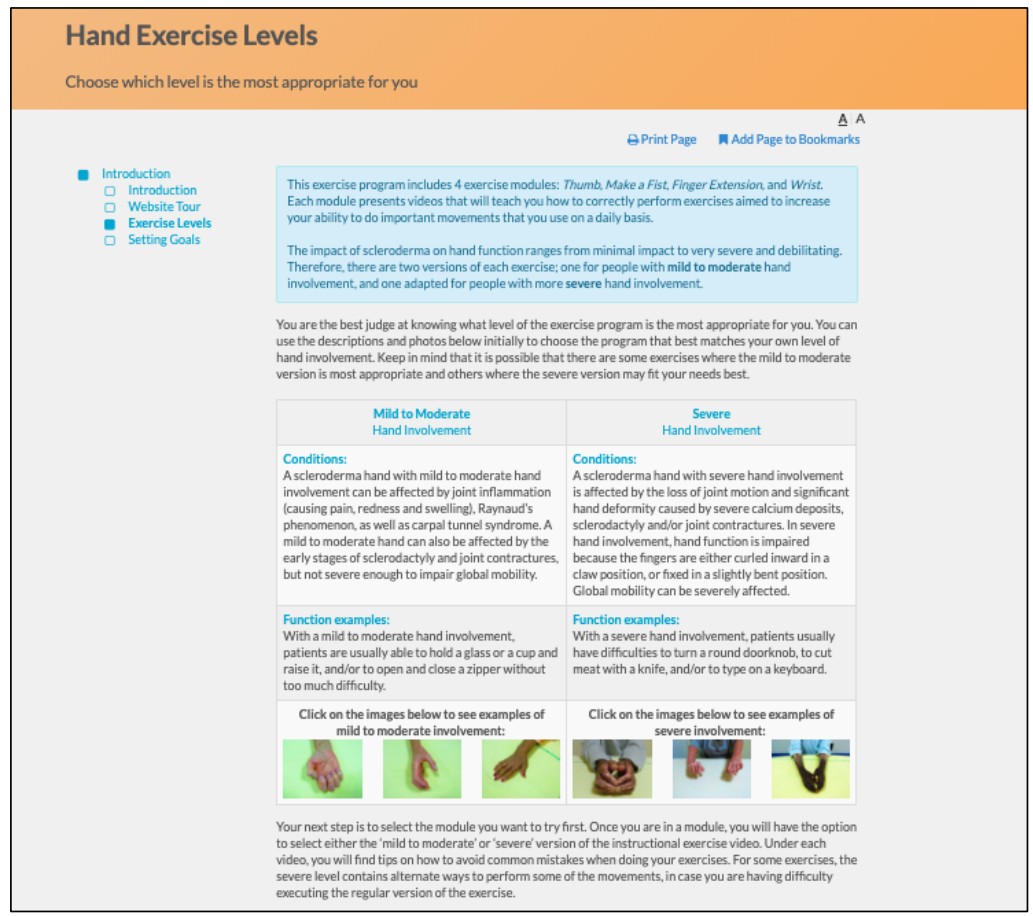

**Figure 5** Information page providing guidance on selecting intervention intensity levels (mild to moderate and more severe hand involvement).

practicalities of delivering the SPIN-HAND Program, recruitment, and uptake, we included a total of 40 SPIN Cohort participants in this feasibility trial.

## Statistical analyses

Descriptive statistics were computed to characterize the sample, as well as for feasibility outcomes including participants' eligibility and recruitment numbers and the percentages of participants who responded to follow-up measures. The frequency of logins and time spent on the SPIN-HAND Program were calculated from the usage log data. Analysis of trial outcome measures included the completeness of data and presence of floor or ceiling effects. Descriptive statistics were used to provide means and standard deviations for the measures, and the standardized mean difference effect size (Cohen's d) was calculated between groups at 3-months follow-up. Qualitative information on participants' experience using the SPIN-HAND program was used to interpret acceptability related to content, webpage visuals, and navigation.

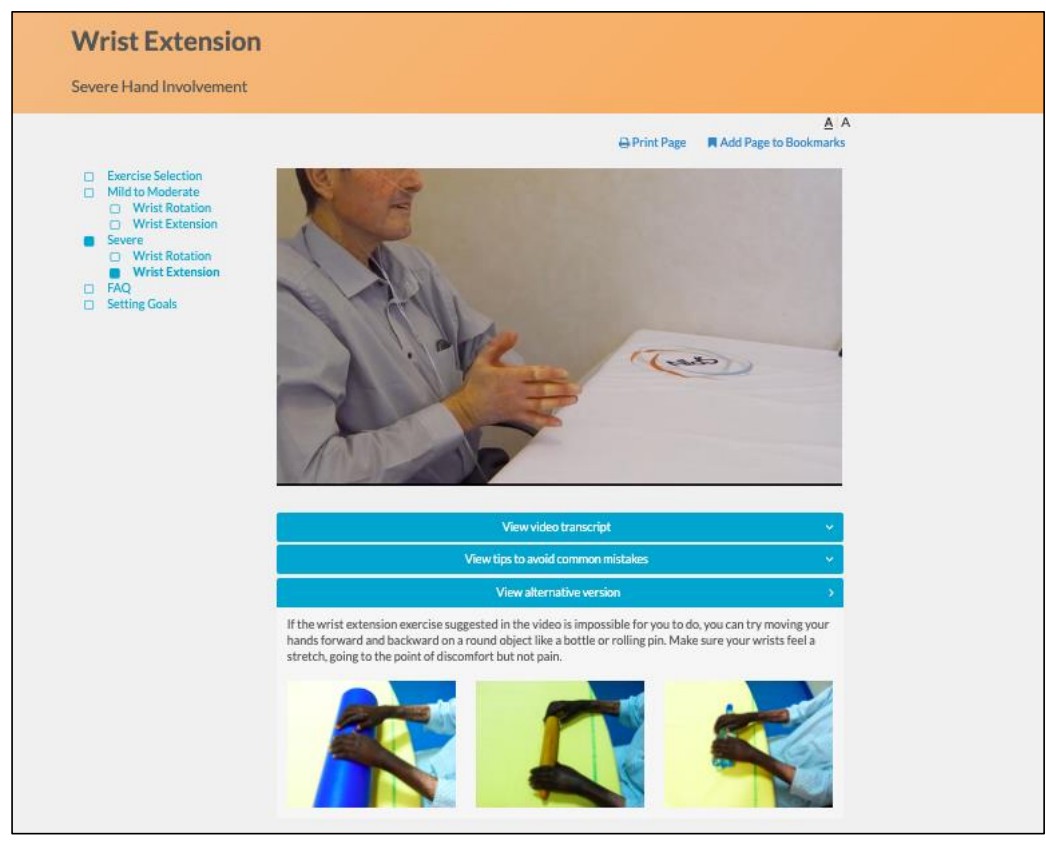

**Figure 6** Pictures illustrating alternate versions on how to perform the exercise for patients with very severe hand involvement.

# RESULTS

## Participant characteristics

Enrollment in the feasibility trial started on June 1, 2017 and was completed on June 18, 2017 when 40 eligible SPIN Cohort participants were randomized. Of these, 24 (60%) were allocated to the SPIN-HAND arm, and 16 (40%) to the usual care arm. See Fig. 7 for SPIN Cohort patients flow through the SPIN-HAND feasibility trial. Demographic and disease characteristics for both groups are shown in Table 1. Characteristics of participants assigned to the intervention and control groups were generally similar with exception of the Modified Rodnan Skin Score, which was substantially higher in the control group.

## Feasibility outcomes

During the period of enrollment, 112 SPIN Cohort participants completed the eligibility forms as part of their regular assessments. Of these, 40 (36%) SPIN Cohort participants met inclusion criteria. The automated eligibility and randomization procedures functioned properly: there were no eligible participants who were not randomized, nor were any non-eligible participants enrolled in the feasibility trial. In total, 15 of 24 (63%) eligible participants randomized to be offered the SPIN-HAND intervention consented to use

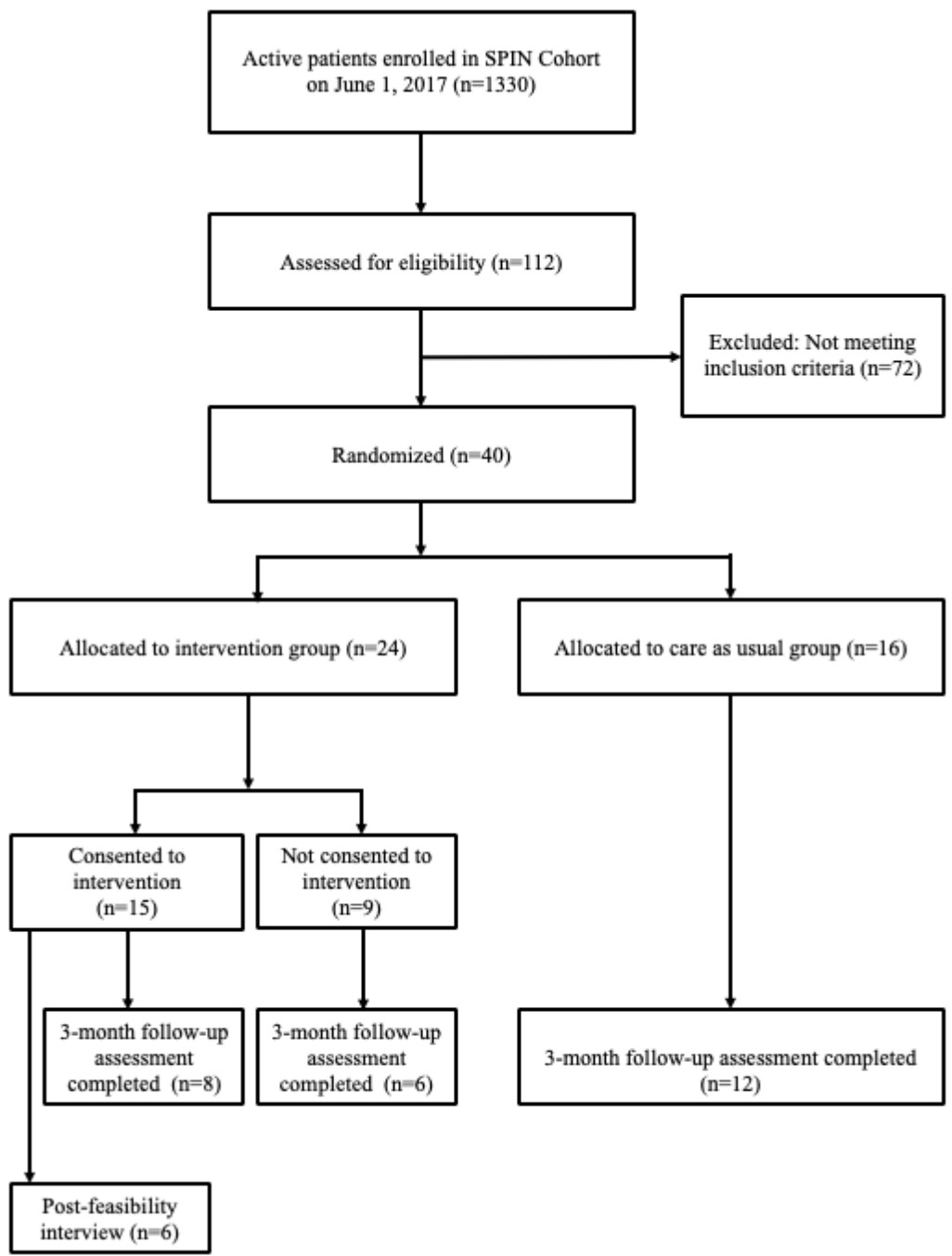

**Figure 7  SPIN Cohort and SPIN-HAND feasibility trial flow.**

**Table 1 Demographic and disease characteristics (N = 40).**

| Variable | SPIN-HAND N = 24 | Usual care N = 16 |
|---|---|---|
| **Demographic** | | |
| Age in years, mean (SD) | 57.1 (13.5) | 58.7 (16.8) |
| Female sex, N (%) | 21 (87.5%) | 15 (93.8%) |
| Education in years, mean (SD) | 15.0 (3.2) | 15.2 (3.1) |
| Married or living as married, N (%) | 17 (70.8%) | 10 (62.5%) |
| Race/ethnicity, N (%) | | |
| White | 22 (91.7%) | 12 (75.0%) |
| Black | 1 (4.2%) | 2 (12.5%) |
| Other | 1 (4.2%) | 2 (12.5%) |
| Country, N (%) | | |
| Canada | 8 (33.3%) | 4 (25.0%) |
| United States | 9 (37.5%) | 7 (43.8%) |
| United Kingdom | 7 (29.2%) | 5 (31.3%) |
| **Disease characteristics**[*] | | |
| Time since onset first non-Raynaud's symptom or sign in years, mean (SD) | 11.8 (7.0) | 11.6 (9.5)[a] |
| Time since onset Raynaud's in years, mean (SD) | 14.6 (11.2)[b] | 15.7 (13.6)[c] |
| Time since diagnosis in years, mean (SD) | 10.0 (6.2) | 10.1 (8.3)[d] |
| Diffuse disease subtype, N (%) | 11 (45.8%) | 8 (50.0%) |
| Modified Rodnan Skin Score, mean (SD) | 9.7 (10.8)[e] | 17.4 (10.8)[f] |
| Small joints contractures, N (% positive) | 6 (25.0%) | 7 (43.8%) |
| Large joint contractures, N (% positive) | 4 (16.7%) | 4 (25.0%) |
| Tendon friction rubs, N (% positive) | 8 (34.8%)[g] | 4 (36.4%)[h] |

**Notes.**
[*]An asterisk (*) indicates that disease characteristics were recorded at time of enrolment in the SPIN Cohort.
Due to missing data:[a]N = 14, [b]N = 23, [c]N = 14, [d]N = 15, [e]N = 20, [f]N = 11, [g]N = 23, [h]N = 11.

the SPIN-HAND intervention. Data coming from the SPIN-HAND platform and SPIN Cohort platform were successfully linked through the unique SPIN ID number assigned to participants 100% of the time.

The total time spent by SPIN personnel on calls and emails for the 24 participants in the intervention arm participants was 307 min (mean = 12.8 min). This included speaking with participants, leaving voicemails, sending emails, and solving access issues. Four calls were dedicated to troubleshoot password issues and difficulty accessing the program. All four calls were resolved by SPIN personnel in less than 15 min, after which participants were able to consent and access the SPIN-HAND program.

Study personnel spent 558 min on tasks related to the feasibility trial other than contacting participants. These tasks involved overseeing the randomization process, tracking progress of enrolment and consents, and logging calls in the SPIN Cohort platform. Between June 1, 2017 and July 17, 2017 (33 working days), which is when the last task (other than calling patients) was completed by SPIN personnel, the mean time spent

on these tasks was 17 min per working day. No issues were reported with the technological performance of the online SPIN-HAND program.

## Usage of the SPIN-HAND program

Of the 24 participants randomized to the intervention, seven (29%) did not log in to the consent page; two (8%) logged in but did not consent, and 15 (63%) consented. Of the 15 who consented, seven (47%) logged in only once, three (13%) logged in twice, and the other five logged in three to 22 times (mean = 4.0; standard deviation = 6.5; median = 2).

After consent, 11 of 15 participants (73%) watched the expert introduction video, and 11 participants watched the patient introduction (73%). Six participants (40%) used the website tour. Ten participants (67%) accessed the page explaining the level of hand involvement (mild/moderate or severe) to identify the exercises most relevant to them. Six participants (40%) did not access any module in the SPIN-HAND Program, four participants (27%) accessed one of the four modules. Two other participants accessed three and four modules (overall mean = 1.1, standard deviation = 1.2, median = 1). Of the different modules in the SPIN-HAND Program, seven participants (47%) accessed the Thumb Flexibility and Strength, five (33%) accessed the Finger Bending module, and four participants (27%) accessed module on Wrist Flexibility and Strength. The module addressing Finger Extension was accessed by one participant (7%). Three of the nine participants who accessed any module watched 1 exercise video, one participant watched two videos. The other participants each watched between three and six videos (mean = 1.8, standard deviation = 2.1, median = 1). Four patients used the goal-setting feature (27%).

## User feedback

At 3-months post-randomization, qualitative semi-structured interviews were conducted with participants in the intervention arm to assess user acceptability and satisfaction. Of the 15 intervention arm participants who consented to use the SPIN-HAND Program, six participated in the interview. The interview questions addressed topics related to usability, understandability, organization, performance and clarity of the SPIN-HAND program. Of the nine participants who were not interviewed, one withdrew a few days after consenting to the SPIN-HAND feasibility trial, three were unreachable at 3-month post-randomization (three contact attempts), and five declined (two participants indicated that they did not use the program enough, two declined due to health problems, and one mentioned that exercise was not a priority). The six interviewed participants were all female with a mean age of 62.3 (SD = 6.7) years. Four had limited disease subtype, the other two had diffuse disease. The mean time since the onset of the first non-Raynaud symptom was 11.2 (SD = 6.6) years, and the mean CHFS score for these participants was 20.2 (SD = 11.0).

A summary of responses to the PEMAT interviews is shown in Table 2. Overall, feedback was very positive. The overall mean grade given by participants for the SPIN-HAND Program was 8.5/10. No concerns related to adverse events were reported.

## SPIN-HAND planned trial outcome measures

Of the 40 participants, 26 (65%) completed their 3-months follow-up assessments, including 14 in the intervention arm (58%) and 12 in the control arm (75%). Table 3

**Table 2**  Summary of responses to the patient education materials assessment tool for audiovisual materials (PEMAT) interviews.

| PEMAT Item | Summary of Responses |
| --- | --- |
| **GENERAL** | |
| Did you use a computer or tablet or both to access the SPIN-HAND Program?Can you please tell us about your experience with the SPIN-HAND Program, including things that you liked about the program and things that could be improved? | 3 computers, 3 tablets. 1 works well, no issues; 1 nothing to improve; 1 likes expert and patient videos; 1 likes exercise levels choices, internet connexion issues; 2 no answer. |
| **PROCESS** | |
| Did the initial invitation email provide you with the information you needed to understand how to sign up for the study? *If no, what information was missing?* | 6 yes. |
| Did you find the follow up telephone call you received within 48 h of the invitation email to be helpful? *If no, why not?* | 5 yes; 1 no because never home when SPIN calling. |
| **PURPOSE** | |
| Did you understand the objective of the SPIN-HAND program? *If no, how could the objective be clarified?* | 6 yes. |
| Did you find the information provided in the SPIN-HAND program relevant? *If no, how could the information provided be made more relevant for you or other scleroderma patients?* | 6 yes. |
| **WORDS AND LANGUAGE** | |
| Did you find that the intervention used common, everyday language that was easy to understand? *If no, can you give an example of something or some word(s) that you did not understand?* | 6 yes. |
| Did you understand all the medical terms or, if not, were they clearly explained in the SPIN-HAND program? *If no, can you give an example of medical term(s) that you did not understand?* | 6 yes. |
| **CONTENT, ORGANIZATION, NAVIGATION** | |
| Did you find that the SPIN-HAND program is broken down into manageable chunks or sections? *If no, which parts of the content weren't broken down into manageable chunks or sections and how could we improve them?* | 5 yes; 1 has not used the program enough to answer. |
| Did you find the different pages or sections of the program to be clearly indicated? *I f no, what section(s) could be more clearly labelled?* | 6 yes. |
| Did you find it easy to navigate through the intervention and to understand where to go next? *If no, how could the different steps to navigate the intervention be more clearly explained?* | 6 yes. |
| Did you consult the "More info" tab (Scleroderma and your hands, FAQ, Patient stories)? *If no, why not?* | 4 yes; 1 only the first time but not on a regular basis; 1 felt overwhelmed with content and hasn't had a chance to look at it yet. |
| Did you experience any technical difficulties while using the intervention? *If yes, what type of technical problems? Did you request assistance from the SPIN team? If you did, was the SPIN team able to help you resolve them?* | 4 no; 1 internet connectivity issue; 1 login issue resolved by SPIN team. |
| Did you use the website tour? *If yes, was it helpful to learn to navigate the website? Why or why not?* | 3 yes, very helpful; 3 no. |
| Did you use the "My bookmarks" feature? *If yes, did you find it helpful for easily navigating to the pages you wanted? Why or why not?* | 6 no. |

**Table 2** (*continued*)

| PEMAT Item | Summary of Responses |
|---|---|
| **VISUAL AIDS** | |
| Did the fact that the intervention was introduced by scleroderma experts and patients make the program more relatable? *Why or why not?* | 4 yes more relatable; 1 yes interesting to hear about others challenges; 1 yes better understanding of symptoms coming from medical professionals. |
| Did you understand how to correctly perform the exercises from watching the videos and listening to the audio instructions? *If no, what would have helped you better understand how to correctly perform the exercises?* | 6 yes. |
| Did you take a look at the "Tips to avoid common mistakes" sections? *If yes, did the pictures of common mistakes and written instructions help you to avoid performing wrong movements? If no, why didn't you use the section on common mistakes section?* | 3 yes very helpful; 3 no (no time, no problem performing exercises, would have looked if need be). |
| Were you able to clearly understand the people speaking in the videos? *If no, why couldn't you understand the words in the videos? (e.g., too fast, too soft, mumbling, accent)? Are there any videos in particular that were more difficult to understand than others? If yes, which one(s)?* | 6 yes. |
| Did you look at the video transcripts? *If yes, were they helpful? If yes, were the video transcripts helpful to you? Why or why not?* | 3 yes very helpful; 3 no. |
| **ACTIONABILITY (Routine, Goal-setting, motivation)** | |
| Did you set an exercise routine for yourself? *If yes, did you find it easy to set an exercise routine for yourself using the materials in the SPIN-HAND program? If no, how could the step-by-step approach be improved or better explained?* | 4 yes it was easy to set a routine; 1 no lack of time, 1 no motivation issue, nothing to improve with step-by-step approach. |
| Did you find an exercise routine that fit your ability level and needs? *If no, what made it hard for you to find an exercise routine that fit your ability level and needs? (e.g., levels not appropriate, time spent on exercises per day or per week not appropriate, other reason)* | 6 yes. |
| Did you set goals for yourself using the goal setting material? *Why or why not?* | 3 yes (very helpful, good daily reminder, did not specify why); 3 no (knew they wouldn't keep up with it, set personal goal but not using SPIN goal form feature, did not specify why). |
| Did you incorporate exercises into your planned routine and stick to it? *If no, what were some obstacles you faced when trying to incorporate the exercises into your routine? How could the SPIN-HAND program have helped you to overcome these obstacles?* | 5 yes; 1 yes but not as much as they would have liked –lack of time and ulcers. |
| Did you use the option to share your goals with friends and family via email? *If yes, did the option to share your goals with friends and family via email help you stick to your goals? If no, what other motivational feature might have been more helpful?* | 5 no - no other suggestion; 1 no - adding testimonials that show the improvement in hand function from people who used the program for a few months. |
| Did you set email reminders for yourself? *If yes, did having the option set email reminders for yourself help you incorporate the exercises into your routine? If no, did you use another type of reminder to do your exercises?* | 1 yes very helpful; 3 no did no use them or any other type of reminder; 1 no did put a note at the beginning of each week in their agenda; 1 no partner reminded them. |
| Did you use the feature to track your progress? *If yes, did having the option to track your progress week after week encourage you to continue performing the exercises? If no, why not? Did you use any other way to track your progress? If so, what did you do?* | 2 yes - felt encouraged to continue; 4 no (2 use their personal tracking system to track improvements, 1 no time to track progress, 1 did not specify why or how). |

**Table 2** (*continued*)

| PEMAT Item | Summary of Responses |
|---|---|
| **OVERALL APPRECIATION** | |
| How user-friendly on a 0-10 scale (0, being the worst and 10 being the best possible score) would you rate the SPIN-HAND program? | 3 rated 10; 1 rated "8 or 9"; 1 rated 8; 1 rated 4 for the initial login, 8 for the program itself so 6 on average. |
| Would you recommend this program to someone with scleroderma? *If no, why?* | 6 yes. |
| What grade (on a 0-10 scale, 0 being the worst and 10 being the best possible score) would you give the program? | 2 rated 10; 2 rated 8; 1 rated 10 for the intent, and "6 or 7" for the program; 1 no answer. |
| Is there anything you want to give us feedback about that was not included in this interview? | 5 no; 1 consider asking people if they liked doing the program on their own of if they would prefer having a PT/OT to coach them. |

**Table 3** Pre- and post- intervention total scores for the CHFS and PROMIS-29v2 Physical Function domain.

| Measure | Intervention N completed | Intervention Mean (SD) | Controls N completed | Controls Mean (SD) | Standardised Mean Difference Effect Size (95% confidence interval) |
|---|---|---|---|---|---|
| Cochin Hand Function Scale | | | | | |
| Baseline | 24 | 21.9 (15.5) | 16 | 21.1 (16.1) | |
| Month 3 | 14 | 22.0 (15.5) | 12 | 20.9 (14.4) | 0.07 (−0.70, 0.84) |
| PROMIS-29 Physical Function | | | | | |
| Baseline | 24 | 40.7 (8.9) | 16 | 39.4 (6.5) | |
| Month 3 | 14 | 42.2 (8.6) | 12 | 41.1 (6.6) | 0.14 (−0.64, 0.91) |

shows the responses to CHFS and PROMIS-29 Physical Function domain for both groups at baseline and 3-months follow-up, which will be the main primary (CHFS) and secondary (PROMIS-29) trial outcome measures of interest for the full-scale trial. Results for the other PROMIS-29 domains and the EQ-5D-5L are displayed in Tables S1, S2. There were no floor or ceiling effects for the CHFS and PROMIS-29 Physical Function scale at baseline or follow-up. The mean CHFS score among those who consented ($N = 15$; $M = 21.3$, SD $= 14.3$) was similar to the mean of those who did not provide consent ($N = 9$; $M = 23.0$, SD $= 18.2$).

## DISCUSSION

Trial methodology for the SPIN-HAND trial, including the automated eligibility and randomization procedures via the SPIN Cohort platform, functioned properly and participants who used the SPIN-HAND program were satisfied with it. All 40 patients were enrolled within 17 days through the SPIN Cohort. However, the consent rate of participants who were offered the intervention (63%) and the usage of the SPIN-HAND program amongst those who consented was low. In addition, only 58% of patients in the intervention group, and 75% in the control group completed their 3 month follow-up measures through their SPIN Cohort assessment.

The SPIN-HAND exercise program is a self-help tool that may improve hand function in patients with SSc. With respect to overall experience with the program, participants reported that the content was relevant and easy to understand. Overall satisfaction with their experience in the SPIN-HAND Program was rated as 8.5 out of 10 on average, and patients would recommend the program to others with scleroderma. Users did not report technological problems with the program. Only six patients completed the post-intervention interview, however. Thus, the satisfaction estimate should be interpreted with caution, as it could overestimate satisfaction if patients who participated in the interview rate the program more positively than the ones not participating. Feedback from people who declined the offer of the SPIN-HAND intervention or who logged in only once would also have provided additional valuable insights. In the full-scale trial, we will use the Client Satisfaction Questionnaire-8 (CSQ-8), a standardised survey that is used to assess satisfaction with health services, at 3-months follow-up (*Larsen et al., 1979*).

There were 24 (60%) participants randomized to the intervention arm and 16 (40%) to the control arm. We carefully reviewed and verified that the randomization procedure with 1:1 ratio was indeed properly implemented and that the imbalance was not due to a programming error. Simple randomization sometimes leads to differences in the number of participants in different trial arms, particularly when the sample size is small, and unequal group sizes reflect randomness (*Schulz & Grimes, 2002*; *Canedo-Ayala et al., 2020*). The degree of imbalance in our feasibility trial would be expected to occur more than 25% with 40 participants randomized 1:1 to two groups.

The low consent rate amongst the participants who were offered to try the SPIN-HAND program in this feasibility trial was problematic and will need to be improved when assessing the program's effectiveness in the full-scale trial. Since we used a fairly low cut-off threshold for the CHFS to become eligible for the trial, one concern might be that participants with lower scores (*i.e.*, limited hand function problems) may have been little motivated to participate. Most intervention arm patients had high CHFS scores, however, and there was no difference in mean scores between those who consented and those who did not, suggesting this was not an issue.

Based on the findings that the uptake of the offer to try the SPIN-HAND program was low, and the low usage of the program amongst consenting participants in this feasibility trial, we will amend our recruitment strategy prior to the SPIN-HAND full-scale trial to try to increase the proportion of patients who accept the offer and access the program's content. Specifically, we will adjust the information on SPIN-HAND that we make available to patients in the SPIN Cohort, by providing them with brief descriptions and screenshots of the program modules, exercise videos, goal setting and patient stories. After providing this information, Cohort participants will be asked whether they would be willing to try the SPIN-HAND program if offered to them (yes/no). To be eligible for the full-scale RCT, patients will also need to answer 'yes' to this question. In addition, in our invitation email to patients who are randomized in the intervention group, we will be providing a brief screencast video that provides an overview of the features of the SPIN-HAND program, thus allowing them to get an idea of what the SPIN-HAND program entails prior to consent.

In the full-scale SPIN-HAND RCT, we will use a 3:2 randomization ratio to ensure a sufficiently large sample in the intervention arm for dose–response and other secondary analyses, without reducing power substantially. There are currently approximately 1,800 active participants in the SPIN Cohort, and 35% of participants were eligible in this feasibility trial. Thus, it appears feasible to include the required sample of 586 patients in the full-scale trial. Since SPIN Cohort participants complete assessments every 3 months, during which they are automatically screened for trial eligibility, patient enrollment should be efficient and fast.

In the cmRCT design, compared with traditional RCT designs, randomization to the intervention or control arm is conducted prior to obtaining consent for the intervention, thus declining participation happens post-randomization. In intention-to-treat (ITT) analyses, patients who do not accept the offer to try the SPIN-HAND program are included in the intervention arm. This allows estimation of the effects associated with offering the intervention but not intervention effects among those who agree to attempt the intervention (*Pate et al., 2016*). SPIN plans to disseminate its programs post-research publicly via partner patient organization websites. For these users, the most important effect estimate relates to users who desire to attempt the program. Therefore, in the full-scale SPIN-HAND Trial, in addition to the primary intention-to-treat analysis, we will use instrumental variable analysis (also known as complier-average causal effect analysis) to estimate effects among patients who accept the intervention offer compared to similar patients in the usual care group (*Relton et al., 2010*; *Steele et al., 2015*).

Although there are currently few published RCTs using the cmRCT design, the acceptance rate in our study is comparable to the rates of intervention uptake in other trials using the cmRCT design. A study on the effects of the offer of adjunctive treatment provided by homeopaths for patients with self-reported unipolar depression in addition to usual care enrolled in the Yorkshire Health Study found that 40% of participants randomized in the intervention arm took up the offer of treatment and had at least one consultation with a homeopath (*Viksveen, Relton & Nicholl, 2017*). Furthermore, in the Utrecht cohort for Multiple BREast cancer intervention studies and Long-term evaLuAtion (UMBRELLA) FIT trial, of the 130 inactive women with breast cancer who were randomized to the intervention group and were offered a 12-week supervised exercise intervention, 68 (52%) agreed to participate (*Gal et al., 2019*).

When the cmRCT design was introduced in 2010, it was suggested that cohort participants can be presented with a list of possible interventions as part of regular cohort data collection and asked if they would agree to use them if offered, as was done in this feasibility trial. It has been hypothesized that this process would identify the potential accepters in advance and consequently reduce dilution of the intervention effects (*Relton et al., 2010*). The results of our study suggest that despite selecting patients based on their indicated interest on the cohort's signalling item, uptake of the offer to try the intervention was low. This result raises important questions about using signalling items as an eligibility criterion for participation in RCTs conducted using the cmRCT design, and it needs to be carefully evaluated how effective these items are at identifying potential accepters of interventions in advance. Since this SPIN-HAND feasibility trial, with its small sample size,

provides only preliminary evidence, additional RCTs with larger samples using the cmRCT design are necessary to confirm this finding.

The completion rates for the follow-up assessments differed between the trial arms (58% vs 75%) and were lower than would be ideal in the full-scale trial. All participants in the SPIN Cohort receive an email two weeks prior to each follow-up to remind them to go online and complete their forms. On the date of the follow-up assessment, a second notification e-mail was sent to patients, reminding them to complete the questionnaires online. If one week prior to the assessment end date the questionnaire had not been completed yet, a SPIN investigator called SPIN Cohort patients to encourage them to go online and complete the forms. In the full-scale trial, calling trial patients may help to get higher completion rates. On the other hand, one of the advantages of the cmRCT design is that patients in the control arm are not aware that a trial is going on (*Eldridge et al., 2016*) and calling them more frequently than usual may alarm them that they are enrolled in a trial. Thus, this additional call will not be implemented in the full-scale trial, but it will be considered for future trials if outcome completion in the full-scale trial also shows to be lower than ideal.

The present study has limitations that should be considered in interpreting its results. First, the SPIN Cohort constitutes a convenience sample of SSc patients receiving treatment at a SPIN recruiting centre, and patients at these centres may differ from those in other settings. Additionally, SSc patients in the SPIN Cohort complete questionnaires online, which may further limit the generalizability of findings, as all participants already have Internet access and are comfortable using it in a research setting (*Kwakkenbos et al., 2019*). Third, we were only able to include English-speaking SPIN Cohort participants in this study. The reason for not including French-speaking patients is that there was a limited number of French-speaking SPIN Cohort participants at the start of this study, and we wanted to be able to assess feasibility aspects but maximize the number of French participants eligible for the full-scale trial. The SPIN-HAND Program is currently not available in Spanish, meaning we could not include Spanish-speaking participants in the trial. Finally, we did not collect medical data (*e.g.*, presence of digital ulcers) during the time of the trial.

In summary, findings of the present study suggest that trial methodology was feasibly implemented and that the online SPIN-HAND program was user-friendly and acceptable to patients. However, acceptance of the offer and usage of the SPIN-HAND Program were low. Thus, adjustments to the information provided to potential participants, aiming to improve uptake of the intervention, will be implemented before undertaking a full-scale RCT of the SPIN-HAND program to assess effectiveness.

## ACKNOWLEDGEMENTS

The SPIN team is dedicating the SPIN-HAND program to the memory of Dr. Serge Poiraudeau, who led the SPIN-HAND project team, along with Dr. Luc Mouthon. Dr. Poiraudeau cared deeply for the quality of life and well-being of people living with scleroderma, and without his leadership and dedication the SPIN-HAND program would not have been possible.

The SPIN Investigators: Susan J. Bartlett, McGill University, Montreal, Quebec, Canada; Karen Gottesman, Scleroderma Foundation, Los Angeles, California, USA; Laura K. Hummers, Department of Rheumatology, Johns Hopkins University School of Medicine, Baltimore, Maryland, USA; Robert Riggs, Scleroderma Foundation, Danvers, Massachusetts, USA; Shervin Assassi, University of Texas McGovern School of Medicine, Houston, Texas, USA; Andrea Benedetti, McGill University, Montreal, Quebec, Canada; Ghassan El-Baalbaki, Université du Québec à Montréal, Montreal, Quebec, Canada; Carolyn Ells, McGill University, Montreal, Quebec, Canada; Kim Fligelstone, Scleroderma & Raynaud's UK, London, UK; Catherine Fortuné, Ottawa Scleroderma Support Group, Ottawa, Ontario, Canada; Tracy Frech, University of Utah, Salt Lake City, Utah, USA; Amy Gietzen, Scleroderma Foundation, Tri-State Chapter, Binghamton, New York, USA; Geneviève Guillot, Sclérodermie Québec, Longueuil, Quebec, Canada; Daphna Harel, New York University, New York, New York, USA; Monique Hinchcliff, Yale School of Medicine, New Haven, Connecticut, USA; Sindhu R. Johnson, Toronto Scleroderma Program, Mount Sinai Hospital, Toronto Western Hospital, and University of Toronto, Toronto, Ontario, Canada; Maggie Larche, McMaster University, Hamilton, Ontario, Canada; Catarina Leite, University of Minho, Braga, Portugal; Karen Nielsen, Scleroderma Society of Ontario, Hamilton, Ontario, Canada; Janet Pope, University of Western Ontario, London, Ontario, Canada; Michelle Richard, Scleroderma Atlantic, Halifax, Nova Scotia, Canada; Tatiana Sofia Rodriguez Reyna, Instituto Nacional de Ciencias Médicas y Nutrición Salvador Zubirán, Mexico City, Mexico; Maria E. Suarez-Almazor, University of Texas MD Anderson Cancer Center, Houston, Texas, USA; Christian Agard, Centre Hospitalier Universitaire - Hôtel-Dieu de Nantes, Nantes, France; Nassim Ait Abdallah, Assistance Publique - Hôpitaux de Paris, Hôpital St-Louis, Paris, France; Alexandra Albert, Université Laval, Quebec, Quebec, Canada; Marc André, Centre Hospitalier Universitaire Gabriel-Montpied, Clermont-Ferrand, France; Elana J. Bernstein, Columbia University, New York, New York, USA; Sabine Berthier, Centre Hospitalier Universitaire Dijon Bourgogne, Dijon, France; Lyne Bissonnette, Université de Sherbrooke , Sherbrooke, Quebec, Canada; Alessandra Bruns, Université de Sherbrooke, Sherbrooke, Quebec, Canada; Patricia Carreira, Servicio de Reumatologia del Hospital 12 de Octubre, Madrid, Spain; Marion Casadevall, Assistance Publique Hôpitaux de Paris - Hôpital Cochin, Paris, France; Benjamin Chaigne, Assistance Publique Hôpitaux de Paris - Hôpital Cochin, Paris, France; Lorinda Chung, Stanford University, Stanford, California, USA; Chase Correia, Northwestern University, Chicago, Illinois, USA; Benjamin Crichi, Assistance Publique - Hôpitaux de Paris, Hôpital St-Louis, Paris, France; Robyn Domsic, University of Pittsburgh, Pittsburgh, Pennsylvania, USA; James V. Dunne, St. Paul's Hospital and University of British Columbia, Vancouver, British Columbia, Canada; Bertrand Dunogue, Assistance Publique - Hôpitaux de Paris, Hôpital Cochin, Paris, France; Regina Fare, Servicio de Reumatologia del Hospital 12 de Octubre, Madrid, Spain; Dominique Farge-Bancel, Assistance Publique Hôpitaux de Paris - Hôpital St-Louis, Paris, France; Paul R. Fortin, CHU de Québec - Université Laval, Quebec, Quebec, Canada; Jessica Gordon, Hospital for Special Surgery, New York City, New York, USA; Brigitte Granel-Rey, Aix Marseille Université, and Assistance Publique Hôpitaux de Marseille - Hôpital Nord, Marseille,

France; Aurélien Guffroy, Les Hôpitaux Universitaires de Strasbourg - Nouvel Hôpital Civil, Strasbourg, France; Genevieve Gyger, Jewish General Hospital and McGill University, Montreal, Quebec, Canada; Eric Hachulla, Centre Hospitalier Régional Universitaire de Lille - Hôpital Claude Huriez, Lille, France; Ariane L Herrick, University of Manchester, Salford Royal NHS Foundation Trust, Manchester, UK; Sabrina Hoa, Centre hospitalier de l'Université de Montréal –CHUM, Montreal, Quebec, Canada; Alena Ikic, Université Laval, Quebec, Quebec, Canada; Niall Jones, University of Alberta, Edmonton, Alberta, Canada; Nader Khalidi, McMaster University, Hamilton, Ontario, Canada; Marc Lambert, Centre Hospitalier Régional Universitaire de Lille - Hôpital Claude Huriez, Lille, France; David Launay, Centre Hospitalier Régional Universitaire de Lille - Hôpital Claude Huriez, Lille, France; Hélène Maillard, Centre Hospitalier Régional Universitaire de Lille - Hôpital Claude Huriez, Lille, France; Nancy Maltez, University of Ottawa, Ottawa, Ontario, Canada; Joanne Manning, Salford Royal NHS Foundation Trust, Salford, UK; Isabelle Marie, CHU Rouen, Hôpital de Bois-Guillaume, Rouen, France; Maria Martin, Servicio de Reumatologia del Hospital 12 de Octubre, Madrid, Spain; Thierry Martin, Thierry Martin, Les Hôpitaux Universitaires de Strasbourg - Nouvel Hôpital Civil, Strasbourg, France; Ariel Masetto, Université de Sherbrooke, Sherbrooke, Quebec, Canada; François Maurier, Uneos - Groupe hospitalier associatif, Metz, France; Arsene Mekinian, Assistance Publique Hôpitaux de Paris - Hôpital St-Antoine, Paris, France; Sheila Melchor, Servicio de Reumatologia del Hospital 12 de Octubre, Madrid, Spain; Mandana Nikpour, St Vincent's Hospital and University of Melbourne, Melbourne, Victoria, Australia; Louis Olagne, Centre Hospitalier Universitaire Gabriel-Montpied, Clermont-Ferrand, France; Vincent Poindron, Les Hôpitaux Universitaires de Strasbourg, Nouvel Hôpital Civil, Strasbourg, France; Susanna Proudman, Royal Adelaide Hospital and University of Adelaide, Adelaide, South Australia, Australia; Alexis Régent, Assistance Publique Hôpitaux de Paris - Hôpital Cochin, Paris, France; Sébastien Rivière, Assistance Publique Hôpitaux de Paris - Hôpital St-Antoine, Paris, France; David Robinson, University of Manitoba, Winnipeg, Manitoba, Canada; Esther Rodriguez, Servicio de Reumatologia del Hospital 12 de Octubre, Madrid, Spain; Sophie Roux, Université de Sherbrooke, Sherbrooke, Quebec, Canada; Perrine Smets, Centre Hospitalier Universitaire Gabriel-Montpied, Clermont-Ferrand, France; Vincent Sobanski, Centre Hospitalier Régional Universitaire de Lille - Hôpital Claude Huriez, Lille, France; Robert Spiera, Hospital for Special Surgery, New York, New York, USA; Virginia Steen, Georgetown University, Washington, DC, USA; Evelyn Sutton, Dalhousie University, Halifax, Nova Scotia, Canada; Carter Thorne, Southlake Regional Health Centre, Newmarket, Ontario, Canada; Pearce Wilcox, St. Paul's Hospital and University of British Columbia, Vancouver, British Columbia, Canada; Angelica Bourgeault, Jewish General Hospital, Montreal, Quebec, Canada; Mara Cañedo Ayala, Jewish General Hospital, Montreal, Quebec, Canada; Andrea Carboni Jiménez, Jewish General Hospital, Montreal, Quebec, Canada; Marie-Nicole Discepola, Jewish General Hospital, Montreal, Quebec, Canada; Maria Gagarine, Jewish General Hospital, Montreal, Quebec, Canada; Julia Nordlund, Jewish General Hospital, Montreal, Quebec, Canada; Nora Østbø, Jewish General Hospital, Montreal, Quebec, Canada.

### Funding

SPIN, including the present feasibility trial, has been funded by grants from the Canadian Institutes of Health Research (TR3-119192, PJT-148504, PJT-149073) and the Arthritis Society. In addition, SPIN has received funding for its core activities, including the SPIN Cohort, from the Canadian Institutes of Health Research, the Arthritis Society, the Lady Davis Institute for Medical Research of the Jewish General Hospital, Montreal, Canada, the Jewish General Hospital Foundation, Montreal, Canada, McGill University, Montreal, Canada, the Scleroderma Society of Ontario, Scleroderma Canada, Sclérodermie Québec, Scleroderma Manitoba, Scleroderma Atlantic, the Scleroderma Association of BC, Scleroderma SASK, Scleroderma Australia, Scleroderma New South Wales, Scleroderma Victoria, and Scleroderma Queensland. Brett D. Thombs was supported by a Tier 1 Canada Research Chair. The funders had no role in study design, data collection and analysis, decision to publish, or preparation of the manuscript.

### Grant Disclosures

The following grant information was disclosed by the authors:
Canadian Institutes of Health Research: TR3-119192, PJT-148504, PJT-149073.
Arthritis Society.
The Canadian Institutes of Health Research.
Lady Davis Institute for Medical Research of the Jewish General Hospital, Montreal, Canada.
Jewish General Hospital Foundation, Montreal, Canada.
McGill University, Montreal, Canada.
Scleroderma Society of Ontario, Scleroderma Canada.
Sclérodermie Québec, Scleroderma Manitoba.
Scleroderma Atlantic, the Scleroderma Association of BC, Scleroderma SASK, Scleroderma Australia.
Scleroderma New South Wales, Scleroderma Victoria, and Scleroderma Queensland.
Tier 1 Canada Research Chair.

### Competing Interests

Luc Mouthon reported personal fees from Actelion/Johnson & Johnson, grants from LFB, non-financial support from Octapharma, and non-financial support from Grifols, all outside the submitted work. All other authors declare: no support from any organization for the submitted work; no financial relationships with any organizations that might have an interest in the submitted work in the previous three years. All authors declare no other relationships or activities that could appear to have influenced the submitted work.

### Author Contributions

- Linda Kwakkenbos conceived and designed the experiments, performed the experiments, analyzed the data, prepared figures and/or tables, authored or reviewed drafts of the article, program development, and approved the final draft.

- Marie-Eve Carrier conceived and designed the experiments, performed the experiments, analyzed the data, prepared figures and/or tables, authored or reviewed drafts of the article, program development, and approved the final draft.
- Joep Welling conceived and designed the experiments, performed the experiments, authored or reviewed drafts of the article, program development, and approved the final draft.
- Kimberly A. Turner performed the experiments, authored or reviewed drafts of the article, and approved the final draft.
- Julie Cumin performed the experiments, authored or reviewed drafts of the article, and approved the final draft.
- Mia Pépin performed the experiments, authored or reviewed drafts of the article, and approved the final draft.
- Cornelia van den Ende conceived and designed the experiments, performed the experiments, authored or reviewed drafts of the article, program development, and approved the final draft.
- Anne A. Schouffoer conceived and designed the experiments, performed the experiments, authored or reviewed drafts of the article, program development, and approved the final draft.
- Marie Hudson conceived and designed the experiments, performed the experiments, authored or reviewed drafts of the article, program development, and approved the final draft.
- Ward van Breda analyzed the data, authored or reviewed drafts of the article, and approved the final draft.
- Maureen Sauve conceived and designed the experiments, performed the experiments, authored or reviewed drafts of the article, and approved the final draft.
- Maureen D. Mayes conceived and designed the experiments, authored or reviewed drafts of the article, and approved the final draft.
- Vanessa L. Malcarne conceived and designed the experiments, authored or reviewed drafts of the article, and approved the final draft.
- Warren R. Nielson conceived and designed the experiments, authored or reviewed drafts of the article, and approved the final draft.
- Christelle Nguyen conceived and designed the experiments, authored or reviewed drafts of the article, and approved the final draft.
- Isabelle Boutron conceived and designed the experiments, performed the experiments, authored or reviewed drafts of the article, and approved the final draft.
- François Rannou conceived and designed the experiments, authored or reviewed drafts of the article, and approved the final draft.
- Brett D. Thombs conceived and designed the experiments, performed the experiments, analyzed the data, prepared figures and/or tables, authored or reviewed drafts of the article, program development, and approved the final draft.
- Luc Mouthon conceived and designed the experiments, performed the experiments, authored or reviewed drafts of the article, program development, and approved the final draft.

## Human Ethics

The following information was supplied relating to ethical approvals (i.e., approving body and any reference numbers):

The SPIN Cohort study was approved by the Research Ethics Committee of the Centre intégré universitaire de santé et de services sociaux du Centre-Ouest-de-l'Île-de-Montréal (#MP-05-2013-150) and by the research ethics committees of each participating centre. Ethics approval for the SPIN-HAND Feasibility Trial was obtained from the Research Ethics Committee of the Centre intégré universitaire de santé et de services sociaux du Centre-Ouest-de-l'Île-de-Montréal (#2019-1146)

## Clinical Trial Ethics

The following information was supplied relating to ethical approvals (i.e., approving body and any reference numbers):

Ethics approval for the SPIN-HAND Feasibility Trial was obtained from the Research Ethics Committee of the Centre intégré universitaire de santé et de services sociaux du Centre-Ouest-de-l'Île-de-Montréal

## Data Availability

The de-identified individual participant data is available at McGill University Dataverse:

Kwakkenbos, Linda and Carrier, Marie-Eve and Welling, Joep and Turner, Kimberly A. and Cumin, Julie and Pépin, Mia and van den Ende, Cornelia and Schouffoer, Anne A. and Hudson, Marie and van Breda, Ward and Sauve, Maureen and Mayes, Maureen D. and Malcarne, Vanessa L. and Nielson, Warren R. and Nguyen, Christelle and Boutron, Isabelle and Rannou, François and Thombs, Brett D. and Mouthon, Luc. 2022. SPIN-HAND Feasibility Trial Results. DOI 10.5683/SP3/TJKGA2.

## Clinical Trial Registration

The following information was supplied regarding Clinical Trial registration:

NCT03092024.

## Supplemental Information

Supplemental information for this article can be found online at http://dx.doi.org/10.7717/peerj.13471#supplemental-information.

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
