# Peer review of "Randomized feasibility trial of the Scleroderma Patient-centered Intervention Network hand exercise program (SPIN-HAND)"

_PeerJ, doi:10.7717/peerj.13471_

## Round 0.1 · original submission · Major Revisions

Thank you for this interesting manuscript on the Randomized feasibility trial of the Scleroderma Patient-centered Intervention Network hand exercise program. I am a bit confused as to the division of the participants. For the 9 participants who were randomized to the intervention, but did not consent to the intervention, how did they login and access the materials for the intervention without consenting? For table 1, were any of the demographics or disease characteristics statistically significantly different for the 2 groups? For the reference list, more than half of the references were greater than 10 years old. I am sure that there are limited papers for this patient population, but it might be helpful to include more up to date information.

·

Basic reporting

meet the standard. good English.

Experimental design

1. please stated the drop-out criteria for the subjects. clearly define the research question, relevant and meaningful.
2. please explain: how do the authors make sure the correct exercise has been done by participants?

Validity of the findings

1. the data about disease characteristics: almost all standard deviation (SD) is more than half of the mean. Could you please clarify why this happens?
2. the author did not give the post-exercise data of EQ-5D

Reviewer 2 ·

Basic reporting

This feasibility trial attempts to analyze the characteristic of the scleroderma patient-centered intervention network hand exercise program on its usability.
The article seems to be almost easily readable, even though in some parts appears hard to follow the description of the SPIN Cohort and SPIN-HAND Feasibility Trial Participants. The references appear almost all up to date.

Experimental design

About the Keywords, it is preferable to choose between MeSH terms.

Introduction:
- line 186 "Previously, three RCTs...". In the text, there is a description of only two RCTs. But the two articles cited appears as a couple of article on this theme and also not the most recent. Which is the third? Also, more recent examples in the literature should be considered.
- line 200 "... to care as usual...". Is not described what the Authors identify as usual because in most of the cases the rehabilitation management of the hand is absent.

Intervention: the description of the 4 modules should include also the timing that the patient spend on each of them. There is a minimum time?

Validity of the findings

No comment

Reviewer 3 ·

Basic reporting

A loss of 20% or greater to follow-up should be a concern of possibility of bias (Schulz & Grimes, 2002). While Table 1 provided useful information on all participants and confirms the success of the randomization process, the substantial proportion of loss to follow-up means that information presented in Table 1 does not adequately describe the analytic sample nor reflect the comparability of the trial groups. The descriptive statistics of analyzed sample and those who are lost to follow-up should be presented separately.

More details are needed about the SPIN’s hand exercise program in the introduction section, such as how SPIN-HAND program components compared with other existing interventions and routine care, online program versus facility or home-based delivery. Some of the information from ‘background and rationale’ section from the research protocol could be incorporated into the introduction.

Lines 186-189, please be more specific about the methodological shortcomings.

It would be helpful to present, briefly, the characteristics of the 6 patients who participated in the feasibility interview in the results section.

Experimental design

The manuscript reported a pilot study, assessing the feasibility for conducting a full-scale randomized controlled trial of an online intervention of SPIN-HAND Program to improve physical function and HRQoL among systemic sclerosis patients. The authors reported a relatively high overall satisfaction and good technical feasibility among patients who completed post-intervention interview. However, these results must be interpreted in light of low consent rate (60%) among those who randomized to intervention, high attrition (7 out of 15 consented patients did not complete 3-month follow-up assessment), and low utilization rates among consented patients (only 5 logged in more than twice). Although the authors acknowledged these limitations, and proposed ways to improve uptake and utilization in the future study, these numbers appeared to indicate a rather unfavorable perspective for a larger RCT. If patients will have to be willing to try the program to be eligible (lines 489-492), the proportion of the 1800 patients who could meet the eligibility criteria may be much lower than 35% as indicated by the authors. This could be further complicated by the potential high lost to follow-up and low adherence to intervention, problems not uncommon in internet-based interventional studies.

Validity of the findings

Table 3 showed that the mean CHFS score at month 3 for intervention patients was higher than control, and the mean CHFS score increased over the study period. A quick look at the raw data, among the 8 patients who completed 3-month follow-up assessment, only 2 had improved/maintained the same level physical function in the hands over baseline (as assessed by the CHFS), while 6 out 12 in the routine care group maintained or had improved physical function. In another words, the patients in the intervention armed were more likely to experience deteriorated hand function after intervention! These numbers showed no sign of the efficacy of the intervention. According to the effect size and standard deviation from Rannou et al study, the sample size of the present study was clearly insufficient to detect the difference in hand function and other functional health outcomes. The author may want to acknowledge the limitations, remove non-feasibility related outcomes from the manuscript, and focus on feasibility measures.

---

## Round 0.2 · Minor Revisions

Thank you for your updated manuscript. It appears that not all points raised by the reviewers were fully addressed.

Reviewer 2 has suggested that you cite specific references. You are welcome to add it/them if you believe they are relevant. However, you are not required to include these citations, and if you do not include them, this will not influence my decision.

·

Basic reporting

no comment

Experimental design

no comment

Validity of the findings

no comment

Additional comments

revision already made and it can full-fills the requirements

Reviewer 2 ·

Basic reporting

Thanks to the Authors for this improved version of the Paper previously submitted to the Journal.
Based on the answers to the reviewers, I shall raise some hints and questions to the Authors as follow reported.

Experimental design

Clarifications on answers to the Editor comments:
1) "I am a bit confused as to the division of the participants. For the 9 participants who were randomized to the intervention, but did not consent to the intervention, how did they log in and access the materials for the intervention without consenting?", if I correctly understood, this could raise possible ethical problems. We can't conceal the information about the treatment on the basis that the patients don't fall into the treatment group. The patients must have full information about the design of the study in order to express valid informed consent. Please describe better on it.

2) About this point “Characteristics of participants assigned to the intervention and control groups were generally similar with exception of the Modified Rodnan Skin Score, which was substantially higher in the control group.” I think is better to specify it on the limits section of the study.

Clarifications on answers to the first Reviewer:
1) The drop-out criteria appear not yet well clarified.

Clarifications on answers to the second Reviewer:
1) Keywords are useful for the indexing of the article. So, terms not included in the MeSH term list are possible but is necessary that terms should have a probability to come in the mind of the possible readers to write it in the search string.

2) Filippetti M. et al in 2020 published an RCT on the effect of a tailored home-based exercise program in patients with systemic sclerosis (DOI: 10.1111/sms.1370). That study included aerobic, resistance training, and hand exercise. I think that also this study could be included as a rational basis to justify the improvement of knowledge on self-administered treatment protocols on these patients.

Clarifications on answers to the third Reviewer:
None.

Validity of the findings

No comment o it.

Reviewer 3 ·

Basic reporting

N/A

Experimental design

N/A

Validity of the findings

N/A

Additional comments

The authors have addressed my questions sufficiently, I have no further comments.

---

## Round 0.3 · accepted · Accept

It appears that you have responded to all of the reviewers' comments.